# Engaging People and Co-Producing Research with Persons and Communities to Foster Person-Centred Care: A Meta-Synthesis

**DOI:** 10.3390/ijerph182312334

**Published:** 2021-11-24

**Authors:** Beatriz Vallina Acha, Estrella Durá Ferrandis, Mireia Ferri Sanz, Maite Ferrando García

**Affiliations:** 1Polibienestar Research Intitute-Instituto de Investigación de Políticas de Bienestar Social: Edificio Institutos de Investigación, Campus de Tarongers, University of Valencia, 46022 Valencia, Spain; estrella.dura@uv.es; 2Senior Europa S.L.–Kveloce I + D + i: C/Roger de Lauria 10–7, 46002 Valencia, Spain; mferri@kveloce.com (M.F.S.); mferrando@kveloce.com (M.F.G.)

**Keywords:** recruitment, hard-to-reach, co-production, health communication, engagement, participatory research

## Abstract

Introduction: Engagement and co-production in healthcare research and innovation are crucial for delivering person-centred interventions in underserved communities, but the knowledge of effective strategies to target this population is still vague, limiting the provision of person-centred care. Our research aimed to identify essential knowledge to foster engagement and co-production. Materials and Methods: A meta-synthesis research design was used to compile existing qualitative research papers on health communication, engagement, and empowerment in vulnerable groups in high-income countries (HICs) from 2008 to 2018. A total of 23 papers were selected and analysed. Results: ‘Design and recruitment’ and ‘engagement and co-production’ thematic areas are presented considering the factors related to researcher–communities attunement and the strategical plans for conducting research. The insights are discussed in light of the literature. Long-term alliances, sustainable structures, and strengthened bonds are critical factors for producing real long-term change, empowering persons and communities, and paving the way to person-centred care. Conclusions: The enhancement of the recruitment, involvement, and empowerment of traditionally disengaged communities and individuals depends on the awareness and analysis of social determinants, power differentials and specific tactics, and the capacity of researchers and individuals to apply all these principles in real-world practice.

## 1. Introduction

Co-production and engagement of individuals and communities are critical for designing person-centred care, and they are even more crucial if the service users are socially vulnerable groups (e.g., traditionally discriminated or marginalised minorities, including racialised groups, deprived communities, or sexual minorities, among others). Vulnerability is understood as the result of specific socio-economic, demographic, cultural, institutional, spatial, and environmental contexts [1] encompassing the susceptibility to hazards, and then, the diminished capacity to cope and/or to adapt. Thus, “vulnerability” is opposed to “resilience”, or the capacity of social, economic, and environmental systems for coping with hazardous events, disturbances, and adverse events, being able to respond and to maintain, transform, or adapt their basic functions, identities, or structures [2].

Traditionally discriminated, marginalised, or excluded communities are considered vulnerable in this article: the historical power imbalance might diminish the ability of these communities and individuals to cope with adverse events due to structural reasons. Other vulnerability conditions that are included are: (i) rural areas, due to their partial isolation and lack of access to key results [3,4,5,6]; (ii) youngest and oldest persons, due to ageism [7,8]; (iii) persons with diagnosis of mental disorders and people with disabilities, due to structural discrimination (mentalism and ableism), stigma, stereotypes, and problems in accessing regular healthcare [9]; and (iv) impoverished persons and communities, also excluded or at higher risk of social exclusion due to systemic determinants [10]. Considering an intersectional perspective, all these disparities potentially leading to marginalisation, exclusion, and discrimination are acquired in the social interaction and reproduced in such a way that various forms of inequalities operate together, exacerbating each other [11,12].

Besides, this study differentiates between hard-to-reach, hidden, and disengaged populations. Hard-to-reach are those groups that are difficult to involve due to their physical, geographical, socio-economic, or socio-cultural situation. Hidden populations are persons who do not wish to be found (e.g., irregular migrants or drug users) [13], whereas disengaged refers to persons or communities not actively involved or considered in the research, co-design of interventions, or even health education programmes (their degree of reluctance may vary, and refers only to a lack of participation).

Considering the above, co-production and engagement present three challenges in health research and participatory research for person-centred care. Firstly, how to raise the recruitment and participation of disengaged and hard-to-reach communities [14]. Secondly, to which extent research could empower or, on the contrary, disempower individuals and communities [15]. Thirdly, how to engage them, paying attention to social determinants, specific difficulties, and power imbalances within the research context [16].

The community-centric approach, as well as individual and community engagement and empowerment, will be further examined to establish the foundation for obtaining knowledge about the recruitment, retainment, alliance-building, and co-production processes:

The community-centric approach is an integrative process to overcome the limitations of traditional research and biomedical models—such as the immutability of individual risk factors or the restrictive focus—by considering the bio-psycho-social perspective [15]. Community-centric approaches can unveil factors that may condition the acceptance of public health campaigns or behavioural change. Consequently, engagement and empowerment are central concepts.

Engagement is widely used in the literature to refer to the community aiming to reach a long-term alliance [17]. The engagement is related to health outcomes for communities and individuals. It is suggested that involving patients as partners in the long term facilitates patient-centred care delivery [18], helping to ensure that research efforts address relevant clinical questions, through the co-design of the agenda, leading to more adequate research design, which may imply to involve community stakeholders [19]. Community engagement includes patients and service users participating in research and healthcare innovation. It entails a long-term involvement (co-production) and alliance at the community level as a whole, implying a multi-stakeholder approach [20,21,22]. Patients’ knowledge, health conditions, beliefs, and experiences impact their decisions to be engaged in interventions and initiatives and, most likely, in research projects [23]. At the individual level, engagement infers the involvement and active participation of an individual in the therapeutical or research process integrating information, professional advice, personal needs and preferences, and their own competencies and abilities in managing their own health [24]. It may also involve a medium- and long-term sustainable relationship with feasible changes at the behavioural level, specifically in self-caring and self-management of health. Moreover, it contributes to the maintenance of health-promotion initiatives by fostering its sustainability over time [25].

Empowerment was understood in this study by considering three key elements: (i) counter-hegemonic rupture; (ii) advocacy; and (iii) locus of control. At the community/social level, empowerment sometimes involves a rupture of the hegemonic order—or the status quo—in which counter-hegemonic forces/groups dispute their own rights, and thus power, in conflict with the ruling elite/s. This notion, related to the rights and collective identity of discriminated minorities (e.g., ethnic minorities), implies deeper and substantial changes in the societal and systemic order itself [15,26]. At the community level, empowerment as advocacy aims at raising the capacity of individuals and communities to control their circumstances through exercising power framed in collective and collaborative efforts. In other words, and specifically applied to health promotion, it implies ‘identity, knowledge and understanding, personal control, personal decision-making, and enabling other patients’ [27]. Finally, empowerment could have a more individualistic meaning, involving both a sense of agency and locus of control [28]; and also in terms of ‘informed patient’ or ‘reflexive consumer’.

The meta-synthesis, based on the meta-ethnography approach sustained by Noblit and Hare [29]—but with some adaptations to effectively cover, relate, compare, and inter-translate the very large sample of studies further detailed in the section on Materials and Methods—has been extensively used for health, social care, and person-centred care research; in particular, for examining the complex relations between a vast range of social and environmental determinants. The meta-synthesis is a systematic review of primary qualitative studies that also aims at building middle-range theory for constructing new frameworks to inform policies, research, and clinical or assistance practice. It applies to multi-faceted health and social care issues by developing new frameworks, comprising health services, policy analysis, strategies, programme and intervention insights, experiences, and social and systemic determinants [30,31]. It was utilised for acquiring knowledge on decision making and patient–provider communication [28], programmes addressed to women with substance use problems [32], studies of exercise programmes for older persons [33], the exploration of the person-centred care concept [34], the experience of disease and the ethnicity [35], or the peer-based support [36].

This study aimed at unveiling the key mechanisms for conducting research under participatory, inclusive approaches involving vulnerable groups through a critical revision of the current qualitative evidence collected in real research project settings. The specific objectives were: (i) to identify key factors to foster recruitment and engagement; and (ii) to improve co-production processes involving vulnerable groups (including hard-to-reach, hidden, and disengaged populations) to support communities in fostering their empowerment at a collective level.

## 2. Materials and Methods

This research summed up the partial results of a meta-synthesis using a sample of 102 studies, including journal articles, PhD theses, and master’s degree dissertations. All studies included in the present article were scientific peer-reviewed articles.

### 2.1. Search Strategy

The literature search was conducted in the University of Valencia database (Trobes) and Google Scholar. Trobes includes the Cochrane Library, SCOPUS, Web of Science, Journal Citation Reports, M3edLine, Proquest Central, and Proquest Dissertations and Theses, among others, and can be found at https://trobes.uv.es/ (accessed on 22 November 2021). Google Scholar was used to find peer-reviewed grey literature, avoiding publication bias. The following keywords were used in both databases:Qualitative studyHealth communicationRecruitmentEmpowermentEngagementChronicLong-term conditionParticipationVulnerableRecruitmentHard-to-reach

### 2.2. Selection Process and Eligibility Criteria

As Figure 1 details, initial results showed 4026 references (3559 after deleting duplicates; 2729 after deleting irrelevant and non-related results) from 2008 to 2018, comprising the period of the economic and financial crisis. Since financial crises imply limited resources devoted to health promotion, investigating how to foster health communication, literacy, self-management, empowerment, and co-production in a period of crisis has a clear intrinsic interest.

Papers meeting the exclusion criteria were excluded from the analysis (see Table 1). As a result, 458 studies were scanned and assessed for the final selection, considering the inclusion criteria and the quality control through the Critical Skills Programme (CASP) questionnaire for qualitative research papers. Finally, 102 studies meeting all inclusion criteria and rated as excellent (CASP ≥ 8) were included in the final sample.

### 2.3. Data Extraction and Analysis

PDFs (full texts of each article) were downloaded. Both the full paper as well as the section on Results were coded and analysed using MAXQDA (Verbi^®®^ v2018–2020) following a meta-synthesis method [29] based on the grounded theory. The grounded theory analysis is based on the lack of predefinition of the themes extracted: the rationale for this innovative approach is that the widely used principles of grounded theory could be applied to a large meta-synthesis in order to ensure that all meanings and metaphors are well captured, but at the same time, the structure of the information is comprehensive but still manageable in order to generate lines of argument and new models and implications for practice. Specifically, the grounded theory approach facilitated that codes and themes arising from the information and the materials were effectively related to each other, and then inter-translated and compared. The qualitative data was tagged according to those repeated ideas and emerging iterative concepts, following an inductive reasoning led by the materials themselves in order to generate the core lines of argument, and then middle-range theories [37].

The analysis was conducted on 102 selected studies. Among them, 23 studies were directly related to the following codes: ‘recruitment for research’, ‘participating in research’ and ‘researching in the community’. Papers selected are cited within the Appendix B.

### 2.4. Characteristics of the Papers Selected for the Meta-Synthesis

Table 2 presents the studies’ characteristics.

The sample analysed in the 23 studies selected covered qualitative data from 786 individuals. However, the figures behind the data remain unclear, since some research papers conducted a participatory observation of very large samples. The sample remains unclear because the audience of these meetings cannot be easily specified. Besides, 37 reports were included by the researchers, as well as 16 visits to stakeholders, and 10 organisations were studied.

The composition of the meta-sample was as follows:160 healthcare professionals (HCPs) (28 females, 2 males, and 120 non-specified);196 healthcare and social workers (e.g., managers, researchers, social workers, community workers, key informants, business and public administration officers, and policy specialists);3 caregivers/relatives;17 patient representatives;356 patients (195 females, 161 males; 79 non-specified).

It is worth noting that some studies included the general population and a vast range of healthcare users with different socio-economic and socio-cultural backgrounds—that may also entail traditionally discriminated persons and communities. These studies were included due to their conceptual richness and the inclusion of patients from all backgrounds. Table 3 sums up the sampled studies.

## 3. Results and Discussion

Twenty-three (23) papers matching the theme ‘Research’ were identified. The papers were related to:Designing and recruiting in research and healthcare innovation for person-centred care. This theme covered the first stages, from the design to the recruitment of individuals and leaders in the community, and how the plan itself, the participation, and the methods used during these phases could contribute to the improvement of the subsequent stages.Engagement and co-production as a means for empowering persons and communities. This covered the research and co-production conduction and how to engage individuals and engagement; thus, phases covered were implementation and, if applicable, evaluation, until the end of each study.

In addition, two analytic themes were used for classifying the information: facilitators and barriers. The constructed items also were classified as factors linked to ‘community–researchers attunement’ and ‘comprehensiveness of the strategic plans’. Each item contained into these two categories was re-interpreted and summarised to capture the key idea behind the statements analysed to make them operative for constructing middle-range theories (full quotations are shown in the complementary materials).

### 3.1. Analysis of Design & Recruitment Factors

Table 4 summarises the key facilitators and barriers (sources are quoted in lowercase letters to facilitate the distinction between the papers comprising the sample, cited in the Appendix B, and the general sources referenced).

As facilitators, the texts analysed emphasised the importance of preparing the information with adequate and tailored content and format, adapted to the community to be reached during the design and recruitment phases (l)(m)(q)(t). It may also imply the cultural adequacy of materials, contents, and means used for their engagement. Enough information needs to be provided, always in an understandable way (l)(m)(q)(t). Community leaders and civil society organisations (CSOs) should be involved, and they can be essential to lead the recruitment, retention, designing of information materials, and dissemination of resources or the assessment procedures (q)(s)(t). It is also relevant to include in the design process an adequate budget planning, as long as the involvement of communities may imply a different allocation of resources compared with traditional methods (l)(p)(t).

The emerged topics show that a real concern, knowledge, and interest in the community is important during this phase (l)(p)(q). To plan how to establish trust and rapport with the participants and, if needed, leaders, is key to ensure their short- and medium-term retainment (b)(i)(p)(r)(t). The qualitative data also reflect that during the recruitment, it is essential to clarify how the study meets the community needs and preferences (g)(x), and to ensure the relevance by finding practical applications of the research and/or its outcomes (p)(x), In some cases, health education or enhancement of this type of literacy should be provided (g)(x). Questionnaires and tools, if not standardised, may need to be adapted, including the terms, language, and jargon (g). To facilitate recruitment, the research and co-production process should be designed with individuals in a flexible way, ensuring adaptability of schedules, meetings, questionnaires, and any other research resource. (i)(q).

The improvement of the cultural competencies and openness of the researchers (g)(x) has been mentioned as a requirement. In particular, the study (x) includes the confidence in narratives (of the community members) and the acknowledgement of their contributions. Once more, trust in researchers appears as a key factor (x)(v).

Scepticism and mistrust are significant barriers to accessing the sample (p)(v)(x)(c), and the difficulty for reaching and engaging the right persons for the recruitment (q). However, in early stages, the main barriers are given by individual factors, such as a lack of self-confidence (d)(q), feeling triggered by sensitive issues (i), or being afraid about legal consequences (c). The lack of information fosters distrust and imbalance of power (t). Fear to disclose private information also appears (i) and, from the side of patients or community organisations, fear of worsening the current stigmatisation of some persons (p). Lastly, the distrust in some communities limits the recruitment scope (v). Other barriers to consider, depending on the group/s targeted, are a disproportionate vulnerability to punitive legal/policy consequences (the fear of persecution may prevent persons and groups from participating (c)), the fear of being further demonised or stigmatised (p), or the previous reluctance to seek medical care (g).

Administrative barriers, highlighting the socio-political environment, are based on the conflicts around the definition of the research agenda, appearing early in the design, even during the proposal stages (p)(t): the lack of resources and the excessive workload to meet specific patients’ and communities’ needs may be an additional source of conflicting agendas (b)(l)(m)(q). These may also prevent the implementation of interventions, programmes, or research projects (b)(l).

### 3.2. Analysis of Engagement and Co-Production Factors

Facilitators and barriers emerging from the texts’ interpretation concerning the engagement and co-production processes are summarised in Table 5.

Concerning the key elements found relevant in the analysed papers for the engagement and co-production with vulnerable groups in health research, the most important facilitators were the active involvement of healthcare professionals (j)(p)(q)(r)(s), the early involvement in the process of researching and innovating (p)(q)(s)(x), the community and the leadership’s trust in the researchers (c)(o)(p)(r)(t)(v)(x), and the provision of feedback and recognition of the contributors’ role, even by acknowledging the co-ownership if possible (l)(s)(u)(v). One of the most important facilitators was to offer a long-term perspective: firstly, by developing community structures, which often involves a meaningful co-ownership (g)(p)(q); secondly, by acknowledging their contributions (l)(s)(u)(v); thirdly, by ensuring financial transparency (m); and fourthly, by offering long-term support by the research entity to the community or advocacy group (l)(p). Another facilitator mentioned was the sensitivity and confidentiality of the process, a requirement to encourage patients to participate in the discussions and feel confident to talk and explain their views (l)(q)(r)(s)(v)(e). To build trust and rapport among participants, community stakeholders, leaders (if any), and researchers is crucial for retaining persons (p)(s)(x)(c).

The researchers′ cultural and linguistic competencies were also essential. Terms should be adapted during the meetings, as well as in materials, questionnaires, or any other resource provided (l)(p)(x). The researchers′ characteristics were considered relevant: their expertise, availability, and social skills, as well as the reputation of their institution and the resources available, matter in engaging and co-producing with vulnerable groups in research (p)(v).

Critical barriers also were identified in the qualitative research, with special relevance of the lack of trust in the researcher (c)(o)(p)(r)(t)(v)(x), which might be mutual (c)(o)(p)(t). In addition, tokenism and technocratic approaches appeared to prevent adequate engagement and co-production processes (a)(f)(k)(l)(p)(q)(t). Communication problems were also emphasised in various studies (i)(k)(l)(p)(q). The issues reported included paternalistic communication, and dismission or rejection of experiential knowledge as subjective, private, and irrelevant (k); researchers also might have problems in communicating well with patients, being misunderstood (l). In (p) and (l), jargon and non-emotional communication (e.g., a researcher dispassionately talking about his disease), and feeling ignored or unable to contribute to the project (l), including feeling disqualified and not listened to (q), were addressed.

Again, conflicting agendas between communities, researchers, and/or research institutions also play a limiting role in the successful engagement of communities and vulnerable individuals (k)(o)(p)(f). Moreover, the greater complexity of participative approaches in research, such as the co-production, is seen as a barrier. Finally, an additional barrier for the researchers to implement these approaches was their potential perception of lack of rigour, mainly due to two main pre-conceptions: firstly, the perceived lack of knowledge of the community in the scientific and/or health-specific topic (the community contributors are laypersons) (j)(k)(l)(p); and secondly, the involvement of the vulnerable groups could be understood as a loss of neutrality (p)(q)(s).

### 3.3. Discussion

This study identified several facilitators and barriers when promoting engagement and co-production of vulnerable communities to participate in health research and innovation. Since this study targeted the academic community, the Community-Engaged Dissemination and Implementation (CEDI) framework of domains and competencies [41] was also integrated within the discussion and, most importantly, in the generation of models. Besides, the social determinants of health, as characterised by Dahlgren and Whitehead [16,42], for researching and intervening in the community [42] were taken into account. The consideration of social determinants of health determines a typology and process for tackling social inequalities in health research, care and innovation [28]. Concerning participation, the classic Ladder of Citizen Participation [43] also is integrated jointly with more modern approaches for planning and executing patients and public engagement. Appendix A (Figure A1). Graphically shows the relations between ladders of participation [43], community engagement, empowerment, and typologies of health research and health interventions in light of health disparities [16,42].

A summary of this meta-synthesis and of the themes that have emerged is presented, aligned with the CEDI model, as well as the main topics of this study: engagement and empowerment through person-centred health research, innovation, and co-production (Figure 2). In particular, nine domains divided into 40 competencies comprised the CEDI model. These domains were reinterpreted within the most important constructs of this research: relevance (affecting the design and the recruitment and considered from the perspectives of researchers and the views from the communities–five domains), engagement (of communities and/or persons–two domains), and empowerment (which mostly involves co-production with the communities–two domains). Our findings as coded from the papers analysed were discussed accordingly (see Figure 2).

Firstly, concerning the first CEDI domain; i.e., the researchers’ attitude towards the community engagement role in research, the analysis revealed the importance of showing sincere concern and interest in the community (p), with an open-minded, interested, and listening attitude shown by the research team (p). Non-participative traditional approaches are still sustained by the academic and healthcare entities involved in research and innovation. Usually, the process of participation is understood as a process of delivering information and then consultation [44], rather than establishing a more profound relationship and engaging in co-production processes. Sometimes, researchers may want too much control without considering the community organisations as co-owners, leading to reluctancy and making it more difficult to reach these traditionally reluctant communities (p). In addition, technocratic approaches and tokenism are barriers to researching the community, even for recruiting persons [44], as seen in (a)(f)(k)(l)(p)(q)(t), since these approaches represent a negative attitude concerning their capacities and/or skills to actively contribute to health research. The extensive training in quantitative analysis and manuscript writing skills dedicated to community members may be a barrier to the implementation of participatory processes in research [45]. In parallel, the researchers’ perception of lack of rigour (j)(k)(l)(p) or lack of neutrality (p)(q)(s) in co-production processes were also identified as barriers for engaging communities in research.

Secondly, the introspection and openness of the researcher—and thus their ability to examine their preconceptions, self-reflection about their own cultural backgrounds, and pre-conceived notions about communities—while practising cultural humility was also found in this meta-synthesis. Community contributors and individuals may struggle in the research and how it potentially contributes to a further stereotyping or marginalisation (p) or even exploitation (t). The marginalisation of experiential knowledge sustained by the participants and community members sustained in the power imbalance must be considered (k)(p)(t). Conflicting realities between communities, even cultures, may arise that also are related to the requirements and duties of all parties involved (m). For instance, communities might not understand the scientific ‘rhythms’ (m), or culturally and linguistically diverse groups would prefer to engage with a researcher from their own community (o). Structural determinants should be integrated within the research and clinical knowledge, practices, and analysis to avoid revictimization, discrimination, and stigmas. In addition, the ‘cultural trauma’ could be addressed: analysis should be periodically conducted through multi-disciplinary meetings; the research and clinical staff should be committed and aware of these structural determinants; and the wider community needs to be involved [46]. During the research design phase, the active involvement of individuals and communities led to more relevant objectives and person-centred care, integrated solutions, or, at smaller scales, user-friendly and understandable information, culturally-tailored interactions, or a richer interpretation of data; it may also enhance the dissemination of results [19].

Third, the knowledge of the community is crucial. The relationship with the community and the cultural awareness of the research team are also key elements (q)(s). The participation of a vast range of actors, a real power-sharing collaboration in co-production and partnership, bidirectional learning, inclusion within the research protocol, and intercultural competency, especially for underserved populations [47]. The role of patients and stakeholders seems to be critical for identifying patient-centred research agendas, priorities, inputs on research design, and strategies for increasing the participation and retention in trials [48].

When approaching hard-to-reach, hidden, or disengaged populations, mistrust appears (p)(v). Researchers may need access to a very specific sample through community leaders, who might be fearful of the consequences that giving access may have. There are not universally adopted strategies for recruiting. Most of them are focused on engendering patients and focused on individuals, while researchers working with minorities highlight the importance of involving the community as a whole [49]

The level of communitarian disengagement appears in (c)(o)(p)(r)(t)(v)(x). To overcome these difficulties, a collaborative reflection embracing a sense of team ethos and community cohesion is required [5]. However, even when designing and executing a flawless protocol for ensuring co-production and community empowerment, to conduct participatory research does not lead to equal participation and cannot break the power imbalance between community members [19]. Participatory research and, specifically, CBPR is effective: (i) in increasing the participation of racial and ethnic minority subjects in research. It also showed to be a powerful tool for diminishing health disparities, determined by the involvement and the degree of engagement by communities; and (ii) for validating effective interventions among under-represented populations [45].

It is crucial to determine how to keep persons motivated to raise retention and engagement. Motivations should be based on advocacy (q). However, there is a lack of standard reporting guidelines and guidance regarding engagement of stakeholders and patients, as well as information on how to identify and recruit participants [48].

The disconnection between the community and the research entity (m)(p) must be prevented for an effective co-production process, and this is particularly important regarding transparency (p)(t). This disconnection might be produced because researchers or institutions are ‘outside of the orbit’ of the community, or if a new programme derived from the research is introduced into the quotidian practice of community workers and community boards (p), among other reasons. The recruitment of participants conducted by community members and leaders increases the recruitment rates and the retainment during the participatory research process [45].

Researchers must be willing to share knowledge with the community to learn about the community and the stakeholders involved, mapping representativeness, leaders, relationships, historical events, power dynamics, or customs to perform a realistic needs assessment focused on them and their priorities [44,45,50,51]. This appeared in (l)(p)(q). To know the community by showing real interest is key before selecting which means would work [13]: for instance, knowledge sharing activities with the communities are critical to ensure a truly participatory approach [45,51]. The role of patients and stakeholders seems to be critical for identifying patient-centred research agendas, priorities, and inputs on research design and strategies for increasing the participation and retention in trials [48]. Well-balanced and equalitarian partnerships in research involve users in all stages and focus on experiential knowledge and mutual learning, with openness and respectful attitudes from both sides [52]. Community empowerment may consist of skill-building, priority setting, action plans’ strategical design and implementation, evaluation, managing budgets or raising funds, or making local management decisions [50].

At the individual level, to directly appeal to patients’ empowerment or participatory methods and co-creation frameworks would help to maximise the recruitment (q)(v).

Fourthly, the appreciation of the communities and individuals’ experiences is essential. The fear of further marginalisation appeared in (o)(p)(t), and trust of community key stakeholders and leaders was emphasised in (p)(s)(x)(c). In this concern, negative previous experiences in research have an impact on their attitudes, and therefore in their involvement in research processes (p)(t)(x).

The fifth domain is to prepare the researchers–community partnership for a collaborative effort, thereby facilitating dialogue, coordination, and decision making. To integrate the formal and informal processes in the community for decision-making, collaboratively outlining the priorities and obtaining the commitment of organisations and leaders for the co-production process is crucial. These factors were mentioned in (r)(s)(t). Researchers should share results while focusing on contextual issues and avoiding re-stigmatisation of service users (p)(x). Both communities and patients need to be heard, valued, and seen (u). In the case of disengaged and highly stigmatised communities, two additional factors are essential for conducting ethical and safe research: (i) to build rapport; and (ii) to consolidate meaningful and safe relationships with the research team (c). A broad range of parties involved is required, including, but not restricted to, community leaders, advocates, community health workers, individuals, health professionals, public administration officers, and service users (e.g., patients). These groups should be carefully defined before designing the strategy for disseminating and implementing the co-production processes [20,22,47]. Activities for sustaining the community engagement should be demarcated as well, and these should include plans far beyond the mere passive receptiveness of information. It is also crucial to share results and provide relevant outcomes, considering their practical application or their potential for generating new resources in the community studied (g)(p)(x).

The multi-stakeholder approach (l), the need for putting in place formal agreements (m), to raise the potential for dialogue and shifting thinking in all parties involved (e), and collaborative efforts and transparency (m) were confirmed as facilitators of the recruitment in this meta-synthesis. Research showed that the motivation and disposition of community members to take part and engage in participatory processes were personal and professional growth, recognition and respect, sense of ownership and sense of confidence, development of leadership skills, knowledge acquisition, concurrence with prevalent cultural norms, appropriateness of the participatory process with local environment and needs, and perceived or experienced beneficial outcomes [50].

The co-production can minimise the power imbalances between researchers, communities, and institutions [44,48,50], as shown in (p)(r)(s)(t). The engagement and outcomes related to participation seemed to be greater for researchers willing to conduct their studies under a multi-disciplinary approach (s). This implied sharing results as soon as possible with community members (p)(r)(t) and providing ownership to communities themselves (r)(t).

All these factors under the fifth domain were also related to the sixth domain, which is the collaborative planning for the research design and goals. The involvement of the populations in the design phase [44,48] since the early beginning (l)(p)(q), as well as the importance of a jointly defined research agenda, were key facilitators (k)(o)(p)(f).

Seventh, the communication effectiveness is related to a clear presentation of ideas, listening attitude, and culturally sensitive and plain language. Our research showed the importance of the information being specifically prepared for the community, culturally tailored, from the design phase (l)(m)(q)(s)(t), using adequate terms (g)(p)(q); and how unclear information (t), jargon (q), or language barriers are problematic in these participative research processes. In sum, communication problems appeared as major barriers for researching involving vulnerable persons (i)(k)(l)(p)(q). Thus, sensitive communication when addressing the problems faced by participants and/or by communities is needed: communication problems in this regard arose as an important barrier [44].

The eighth factor is the equitable distribution of resources and credit, including financial resources, resource distribution, and role in media coverage and scientific publications. Planning how to share results (p)(r)(t) and provide ownership (l)(r)(s)(t)(u)(v) appeared. In addition, budget and resources were mentioned (r)(e), as well as financial transparency (m). In this regard, it is worth mentioning that collection, interpretation, and exchange and co-ownership of data are very complex issues that could be discussed in terms of the consequences of these processes for the groups themselves. Data gathering may be detrimental to deprived or marginalised communities. Besides, the indicators may be culturally inadequate, the stakeholders may be under-represented or disregarded, and the results are sometimes inaccessible for them [15,27].

Lastly, the ninth domain is the long-term relationship, sustaining the partnership and integrating the capacity building, the long-term funding, the self-sustainability of research and healthcare innovation projects and programmes, and the commitment of devoting time and efforts to addressing stakeholder needs beyond the research [52]. Long-term financial and organisational support (l)(p) appeared jointly with results sharing (q)(s)(t). Both factors seem to be linked to the dependency on funds by patient organisations (l) and the lack of long-term engagement of researchers perceived by the community (m). Organisational commitment and policies, budget, resources, time, and training (r)(e) are important for preventing a negative sense of exploitation (p)(t)(x) or abandonment (o). To overcome these difficulties and perceptions, structures to sustain the communities’ empowerment in the long-term could be provided (g)(p)(x) while also considering the sharing of resources (p)(t) and raising policies and organisational commitment for ensuring the persons’ participation (r)(e).

The goal of our study was to provide sound knowledge for research into person-centred care by empowering deprived communities to be part of research and innovation in health and social care. To achieve this empowerment, we must integrate the lessons learned and the findings exposed in these results, as well as the ladders of participation [43] and the typology of actions on health inequalities [42].

#### 3.3.1. Implications for Practice

The implementation of co-production methods during the entire research process could offer a more tailored process and a more successful recruitment, raising the recruitment and retainment of traditionally disengaged communities. It is needed to engage participants, advocacy entities, and groups in the long term.

The checklist in Figure 3 summarises our findings translated into best practices, divided into the phases of the design and execution of a person-centred co-production.

The role of the research and the co-production for studied communities depends on their capacity to produce a real and long-term change that translates into meaningful empowerment of persons and communities, therefore paving the way to more efficient person-centred care. There are some important considerations when fostering person-centred care through the engagement and empowerment of individuals and communities.

On the one hand, at the individual level, to offer flexibility and to foster self-efficacy to ensure safety and confidentiality are key aspects. During the co-production, to engage participants, raise their confidence, adapt terms, and overcome the tokenism are essential to empowering vulnerable persons. Likewise, to prepare appropriate information addressed at individuals and communities to be involved, founded on a sound knowledge of the community, is needed. On the other hand, at the community level, it is required to consider the disengagement, the need for structures, and long-term alliances to empower communities at practical and real levels. Vital factors include tackling the power imbalance during the early stages, involving health professionals and key stakeholders during the execution of co-creation processes, and building trust. Trust and acknowledgement of community contributors are decisive during the design and recruitment stages, as well as during the execution phases. The acknowledgement of their contributions, even co-ownership (whenever possible or pertinent), should be considered.

#### 3.3.2. Limitations and Strengths

The great amount of literature reviewed, together with the qualitative perspective selected to analyse the discourse of key persons involved in the research, constituted a strength of the current research. Qualitative research allowed us to obtain a vast range of information about the participants’ perspectives, solutions proposed, assumptions, beliefs, strengths, and structural problems in their own terms, which is crucial for designing well-tailored interventions attending to the targeted audiences’ narratives. Likewise, the integrative perspective and the focus on the individuals’ agency and the community empowerment for focusing the engagement was a meaningful strength of our study. The innovative methodology for analysing a vast amount of data through the integration of grounded theory principles were used, although the general structure was a meta-synthesis due to the huge number of papers and dissertations (102 in total; 23 of them were included in this study) included in the general dissertation. The innovative approach allowed us not to lose important information by: (i) facilitating the extraction of key thematic areas arising from the information; and (ii) not attaching to a single framework (e.g., the CEDI) delimiting the codes and the analysis of the information, but facilitating the comparison and the analysis with the literature.

The study also had some limitations that relied on the non-exhaustiveness of the literature sample included. As a matter of fact, this research was a part of a larger meta-synthesis not entirely focused on research and co-production, but strongly linked to these topics. Conversely, the ‘partial’ nature ensured its integration into a wider frame, joining key themes on health communication and patient and community participation: decision making, self-management of health, or counselling were jointly treated in a comprehensive viewpoint, which supposed a great added value in terms of integrative perspectives able to inform practice.

Further research will be conducted on the prioritisation of factors and on the validation of the checklist and best practices extracted. The outputs will be tested in real research settings through qualitative research to validate and prioritise those that would be specifically needed for person-centred care. The research results will be disseminated to health and social care professional and workers in plain language.

## 4. Conclusions

This qualitative study reviewed and showed the experiences of researchers, patients, and communities, as well as other relevant stakeholders, such as public administration officers. The knowledge captured in these experiences allowed us to identify facilitators and barriers, as well as provide best practices for conducting a culturally tailored and empowering co-production in research and healthcare innovation. Three challenges were addressed in the study design: (i) how to raise the recruitment of the disengaged; (ii) the potential for empowering persons and communities through research and/or co-production; and (iii) the engagement of deprived and disengaged groups in light of the social determinants of health. The co-production during all research phases offered a well-tailored process, and successful recruitment of participants and key stakeholders. The safety of the participants, the confidentiality of the information, and the awareness of power imbalances—both between researchers and communities and among participants—during the research was critical from the point of view of the effectiveness, engagement, and ethics of the study during its entire lifecycle. When designing innovation and/or interventions involving vulnerable communities, socio-economic and cultural settings and systemic co-determinants should be integrated within the co-production process. Collective empowerment depends on the access to basic services (among others, healthcare) and the material circumstances, equitable societal structures (policies, budget and funds, and approach and focus of the measures undertaken by decision makers), and structural determinants (again, socio-economic, cultural, and environmental conditions co-determining health inequalities at a macro level). Co-production processes could be approached for raising the individuals′ self-efficacy, skills, and access to services; the advocacy capacities of the community and civil society players; and for supporting long-term changes and promoting healthier policies able to change current vicious cycles of poverty and poor health outcomes. To collaborate with communities and to sustain or, at least, not interfere in the creation of sustainable structures and bonds will result in meaningful empowerment and an enhanced sense of agency of all participants.

## Figures and Tables

**Figure 1 ijerph-18-12334-f001:**
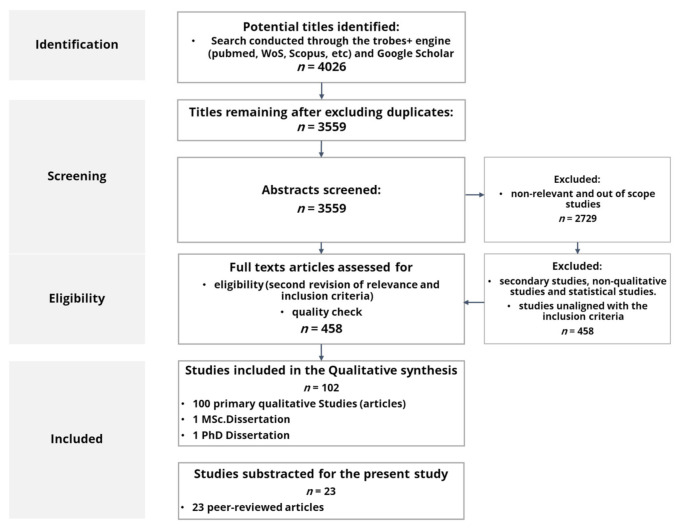
Search strategy and result diagram.

**Figure 2 ijerph-18-12334-f002:**
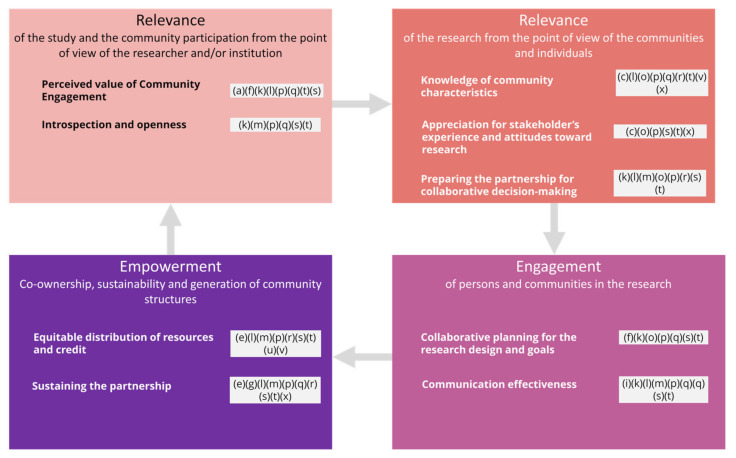
Results discussed in light of the CEDI model.

**Figure 3 ijerph-18-12334-f003:**
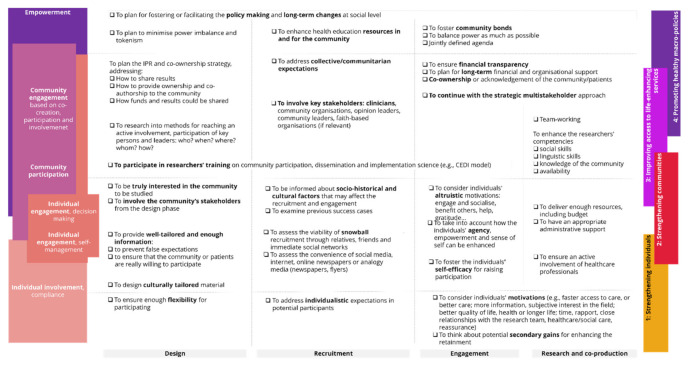
Checklist to promote community empowerment and individuals′ engagement in research for person-centred care.

**Table 1 ijerph-18-12334-t001:** Inclusion and exclusion criteria.

Inclusion Criteria	Exclusion Criteria
☑ Peer-reviewed (articles, PhD dissertations…) ☑ Primary study ☑ Qualitative study ☑ Mixed method study ☑ From 2008 to 2018 (economic crisis and partial recovery) ☑ High-income countries (transferability of results) ☑ Adult populations ☑ In situations of psycho-social vulnerability ☑ CASP ≥ 8	🗵 Conflict of interest declared 🗵 Non-peer reviewed (e.g., reports) 🗵 Systematic reviews and meta-synthesis 🗵 Quantitative studies 🗵 Before 2008 or after 2019 🗵 Low- and medium-income countries 🗵 Focused on children and teenagers; minors 🗵 Focused on older persons

**Table 2 ijerph-18-12334-t002:** Studies included, characteristics, and ethical considerations.

	Year	Type of Study	Ethical Statement and Considerations	
a	2010	Scientific article		Approved by Thames Valley multi-centre research ethics committee.
b	2016	Scientific article		Approved by the Health Research Ethics Board at the University of Alberta.
c	2016	Scientific article		Approved by Institutional Review Boards at the University of California. Ethics issues widely discussed within the study.
d	2016	Scientific article	**x**	The study did not need approval of the ethical review board according to the Dutch Medical Research Involving Human Subjects Act (WMO); only (non-intervention) studies with a high burdenfor patients require a review.
e	2012	Scientific article		Ethics approval was gained from the Joint UCL/UCLH Committees on the Ethics of Human Research.
F	2009	Scientific article	**x**	AN explicit ethics statement was not found.
g	2012	Scientific article		Approved by the Jackson State University Mississippi and the Indiana University Kokomo Institutional Review Boards.
h	2011	Scientific article		Approved by the Eastern Multi-Centre Research Ethics Committee.
i	2012	Scientific article		Approved by the ethics committee of the University of Auckland, through which the research was undertaken.
j	2017	Scientific article		Approved by the James Cook University Ethics Committee, Queensland University, and La Trobe University Faculty of Health Sciences Human Ethics Committee.
k	2008	Scientific article		Ethical considerations were discussed and detailed; approval procedures were not explicitly stated.
l	2013	Scientific article	**x**	The study did not explicit state the ethical procedures and approval.
m	2015	Scientific article	**x**	The study did not explicit state the ethical procedures and approval: each project seems to have obtained the ethical approval from its respective committee.
n	2015	Scientific article	?	The study did not explicit state the ethical procedures and approval: it seemed to depend on the clinical trial registry and approval.
o	2014	Scientific article	**x**	The study did not explicit state the ethical procedures and approval, but offered detailed information about data and research integrity and participatory procedures.
p	2009	Scientific article		Approved by the appropriate institutional review boards (National Institute of Mental Health and the HIV Center for Clinical and Behavioral Studies).
q	2014	Scientific article		This qualitative study formed part of the ‘Evidence base for Patient and public Involvement in Clinical trials’(EPIC) project. The National Research Ethics Service (NRES) advised that EPIC did not require NRES ethics approval; we therefore sought and obtained a favourable ethical opinion from the University of Liverpool Research Ethics Committee.
r	2015	Scientific article	**x**	As this was an independent consultation exercise on behalf of the London Sexual Health Programme, ethical approval was not required.
s	2015	Scientific article		Although the ethical approval was not explicit, a detailed section on quality procedures and ethical considerations was included.
t	2016	Scientific article		The Albert Einstein College of Medicine′s Institutional Review Board approved the study.
u	2013	Scientific article		The regional committee for medical ethics in Central Norway approved the study, and it was registered with the Norwegian Data Inspectorate.
v	2017	Scientific article		The study was approved by the institutional review board at the authors’ institution (Department of Health Education and Behavior, University of Florida).
x	2016	Scientific article		Ethical approval for this project was granted by the Aboriginal Health and Medical Research Council.

**Table 3 ijerph-18-12334-t003:** Thematic coverage and sample of the papers included in the meta-synthesis.

	Year	Geographical Coverage	Sample	Social Factors to Be Considered and Rationale for Their Inclusion	Focus
a	2010	UK	21 patients	The paper aimed at reflecting the maximum variety in age, sex, ethnicity, health literacy, IT literacy, stage, severity of condition, presence of other illnesses, and extent of family support.	PATIENT PARTICIPATION (IT topics)
b	2016	CANADA	32 female HCPs and 2 male HCPs; 8 non-specified; 48 nurses; 8 patients, gender non-specified	Persons with HIV and nurses. Nurse mentees worked in a range of clinical areas, including community or public health, sexually transmitted infection clinics, prisons, long-term care, acute care, and mental health.	HIV
c	2016	USA	33 female patients	Irregular migrant sex workers	HIV
d	2016	NETHERLANDS	29 patients:19 females and 10 males	Patients from several settings and healthcare professionals; most respondents had a low or medium level of education, and the majority of respondents were unemployed.	ARTHRITIS
e	2012	UK	43 HCPs, gender non-specified	Relevant due to its direct link to patient empowerment, covering a vast range of socio-cultural and economic backgrounds. The sample was formed by 43 hospital respiratory consultants, nurses, and general managers at 24 intervention and 11 control NCROP sites.	COPD and ASTHMA
f	2009	USA	100 workshops, 2 organisations, and 5 managers	Psychologists working on human service organizations (non-profit, community-based organizations that provide social and health services to low-income individuals and families living in impoverished, urban communities.)	PATIENT PARTICIPATION
g	2012	USA	20 patients: 14 males and 6 females	Racialised (African-American) men, historically discriminated, with low level of access and use of healthcare resources. Per capita and household incomes, as well as education, were discussed in the sample description.	PROSTATE CANCER
h	2011	UK	42 patients: 27 females and 15 males	The paper reflected a wide variety of conditions and socio-cultural-economic backgrounds, as well as explicit vulnerability issues; e.g., persons diagnosed with severe mental disorders (*).	CHRONIC DISEASES
i	2012	NEW ZEALAND	26 female patients	Women with STI: human papilloma virus or genital herpes simplex virus (**).	VIRAL STI
j	2017	AUSTRALIA	Focused on 45 workshops and 16 visits to stakeholders	The paper included populations from rural settings (***) and from a diverse range of regions.	GENERAL HEALTH
k	2008	UK	18 HCPs, without specifying gender; 2 general practitioners and 16 nurses; 17 patients (no gender specified)	The paper reflected a wide variety of conditions and social backgrounds. It included caregivers and caretakers with serious long-term illnesses (****).	PATIENT PARTICIPATION (patient council)
l	2013	NETHERLANDS, UK	14 researchers, 1 business officer and, 1 policy specialist; 16 patients: 9 females and 7 males	The paper reflected a wide variety of social backgrounds, and all patients suffered from chronic diseases coursing with complicated pain. (*).	PATIENT PARTICIPATION (conferences)
m	2015	USA	25 researchers	Eleven projects focused on largely African-American communities, nine focused on communities of mixed ethnicities, and five focused on immigrant or refugee populations and communities with lower access to healthcare.	COMMUNITIES
n	2015	GERMANY, USA	20 female patients	Women with breast cancer and psychological vulnerability (*), including ethnic minorities, within a clinical trial (further information at https://www.cancer.gov/types/breast/research/star-trial-results-qa accessed on 22 November 2021; https://clinicaltrials.gov/ct2/show/NCT00003906 accessed 22 November 2021).	BREAST CANCER
o	2014	USA	31 patients: 22 females and 9 males	This paper explicitly addressed communities and health disparities, including individuals from a diverse range of backgrounds, including ethnic minorities, different age groups, education, employment status, and income levels.	COMMUNITIES
p	2009	USA	2 HCPs; 2 nurses, gender non-specified; 2 social workers; 10 key informants	10 community-based organisations working on HIV prevention.	HIV
q	2014	UK	38 researchers; 28 reports	Clinical trials. Chief investigators (CIs) and patient and public involvement (PPI) contributors were engaged; thus patients were from multiple sociocultural backgrounds, including dependent patients and caregivers (****).	PATIENT PARTICIPATION
r	2015	UK	5 HCPs (gender non-specified); 1 researcher; 8 health-related professionals (e.g., public health, health promotion specialists, etc.); 3 patient representatives; 2 patients (gender non-specified; they seemed to be males); 8 organisations	Underrepresented communities, and voluntary or community organisations.	HIV
s	2015	NETHERLANDS	13 researchers, 17 other officers (e.g., health funds); 7 public administration officers; 14 patient representatives; 9 reports considered	Maximum variation of participants was ensured in this study, including persons with some vulnerability conditions, such as intellectual disabilities.	DIVERSE CONDITIONS (cord injury; asthma; COPD; diabetes; neuromuscular diseases; renal failure; congenital heart disease; intellectual disabilities; burns)
t	2016	USA	14 community members (genders non-specified)	Minority members of a community advisory board in AIDS/HIV research.	HIV, engagement, and CBPR
u	2013	NORWAY	44 HCPs (gender non-specified); 13 public administration officers; 20 patients (gender non-specified)	Health professionals and patients participated in the study, which covered several backgrounds and, in particular, mental disorders (*).	PATIENT PARTICIPATION
v	2017	USA	60 patients: 33 females and 27 males.	African-American males. Research was conducted in Alachua County, Florida: approximately 25% of the residents lived below the federal poverty level, and African-Americans comprised 20.5% of the population, and had the highest morbidity rates from chronic diseases.	PATIENT PARTICIPATION (IT topics)
x	2016	AUSTRALIA	35 health workers (with HCPs being the vast majority) 31 of which were females; 6 aboriginal health workers; 2 allied health professionals; 6 nurses; 9 managers; 7 doctors; and 5 administrators	Aboriginal health. Health professionals from 2 urban and 1 regional Aboriginal Community Controlled Health Services (ACCHS) in New South Wales	COMMUNITIES

(*) Mental disorders, and in particular highly stigmatised ones (e.g., borderline personality disorder (BPD)), were considered as a vulnerability condition, due to the systemic discrimination that these individuals may suffer, but also because of their lower access to resources. In parallel, mood disorders and important emotional disturbances linked to the chronic condition were considered within these conditions. (**) Sexually transmitted infections (STIs) were considered a vulnerability factor for accessing healthcare and for participating in the research due to the stigma linked to these conditions, mediated by gender expression and sex. (***) Rural populations were considered vulnerable because of the difficulties in accessing healthcare and health literacy, lack of infrastructure, and tele-communications challenges, among other potential difficulties, such as impoverished or deprived socio-economic circumstances. [3,4,5,6]. (****) Caregiving for dependent persons had a direct impact on the household financial burden, as well as very particular employment and workload challenges, while also mediated by other structural factors mainly related to age and gender [38,39,40].

**Table 4 ijerph-18-12334-t004:** Facilitators and barriers during design and recruitment.

**Design and recruitment**	**Facilitators**	**Community–researchers attunement:** • **Communities’ factors:** ○ Information prepared, adequate to the community and or individuals (l)(m)(q)(t), including cultural appropriateness (q)(s) ○ Trust-building actions with the community or the participants (b)(i)(p)(r)(t) ○ Strategies implemented for reducing power imbalance and tokenism (r)(s)(t) ○ Ensured flexibility for participation integrated into the design from the beginning (i)(q) • **Researchers’ factors:** ○ Researchers able to involve the community from the design phase (l)(p)(q)
		**Comprehensiveness of the strategic plans:** • Knowledge and competencies for implementing methods aimed at ensuring an active involvement, and participation of key persons and leaders (q)(s)(t) • **Logistics and institutional factors:** ○ Plan established for sharing results (p)(r)(t) ○ Training foreseen or conducted for researchers in participatory research and co-production of healthcare innovation (p)(s)(t) ○ Training foreseen or conducted for community members and participants involved (m)(p)
	**Barriers**	**Community-researchers attunement:** • **Individuals’ co-determinants:** ○ Individual and personality factors that may undermine the participation, such as the lack of self-confidence (d)(q), or triggered by sensitive issues (i); fear of legal consequences (c) • **Communities’ factors:** ○ Distrust of community leaders and civil society organisations (CSO) based on the fear of more stigmatisation and stereotyping (k)(p)(t) ○ Disconnection between the community and the research entity (m)(p)(o) ○ Lack of reimbursement for community organisations according to their workload (p)(t) ○ Perceived lack of information and lack of transparency (t) ○ Lack of long-term engagement and commitment with the community (m) ○ Conflicting agendas between researchers and communities (p)(t) ○ No training available or conducted for community members and participants (m)(p) • **Researchers’ factors:** ○ Conflicting agendas between researchers and communities (p)(t) ○ Use of jargon, specialised terms, and, in general terms, non-adapted information (q)
		**Comprehensiveness of the strategic plans** • Previous training conducted for researchers in co-production and participatory approaches (l)(q) • Lack of leadership, support, and opportunities in the research institution (b)(q) • **Logistics and institutional factors:** ○ Administrative issues related to workload, allocation of resources (b)(l)(m)(q) ○ Logistical problems in accessing the sample (q) ○ Role conflicts in the same person researching; e.g., clinician and researcher (q) ○ Unaffordable scientific demands such as deadlines, calls, or due dates (m) ○ Lack of financial resources and strong dependency on funds by patients’ organisations (l)

**Table 5 ijerph-18-12334-t005:** Facilitators and barriers during the engagement of participants and the co-production process.

**Engagement and co-production**	**Facilitators**	• **Community–researchers attunement:** • **Communities’ factors:** ○ Individuals encouraged for engaging in discussions, feeling confident to talk and explain their views (l)(q)(r)(s)(v)(e) ○ Trust and rapport built between communities and researchers/institutions (p)(s)(x)(c) ○ Safety, security, and anonymity (i)(c)(r)(v) ○ Meaningful co-production, used for meeting community goals and needs (g)(p)(q) ○ To feel as being heard, valued, and seen (u) ○ To perceive empathy from the researchers (i) ○ To give participants a sense of responsibility (v) ○ To provide opportunity for producing shifts in thinking (e) ○ To conduct collaborative efforts and deliberations (s) • **Researchers’ factors:** ○ Trust and rapport built between communities and researchers/institutions (p)(s)(x)(c) ○ To provide opportunity for producing shifts in thinking (e) ○ To conduct collaborative efforts and deliberations (s)
		• **Comprehensiveness of the strategic plans:** ○ Healthcare professionals actively involved (j)(p)(q)(r)(s) ○ Possibilities for providing co-ownership or acknowledgement (l)(s)(u)(v) ○ Clear expectations about the intervention (l)(p)(q) ○ Terms adapted; e.g., in meetings, materials, questionnaires (l)(p)(x) • **Logistics and institutional factors:** ○ Organisations committed in their own internal policies to the research and/or the co-production process (r)(e) ○ Budget, resources, time, and training available for the research team and the community involved (r)(e) ○ Long-term financial and organisational support for the community (l)(p) ○ To include capacity building in team-working skills (e) ○ Financial transparency (m) ○ Good reputation of the research institution (p)(v) ○ Accessibility of materials, and performing adaptations if needed (l)
	**Barriers**	• **Community–researchers attunement:** ○ Communication problems between communities/participants and researchers (i)(k)(l)(p)(q) ○ Conflicting agendas between the research institution/researchers and the community leadership (k)(o)(p)(f) • **Communities’ factors:** ○ Researchers viewed as outsiders by the participants (c)(o)(p)(r)(t)(v)(x) ○ Lack of trust in the researchers (c)(o)(p)(t)(v) ○ Community’s sense of “exploitation” perceived (p)(t)(x) ○ Notable power imbalance between researchers and communities/individuals (p)(t) ○ Emotionally charged communication and perceived intrusiveness regarding sensitive issues (i)(p) ○ Lack of sense of community and cohesion (c)(o) ○ Sensed ‘abandonment’ of a community/neighbourhood (o) ○ Incompatibilities between the research and the socialisation environment (r) • **Researchers′ factors:** ○ Researchers’ technocratic approaches and tokenism (a)(f)(k)(l)(p)(q)(t) ○ Perceived lack of rigour of the participatory approaches (j)(k)(l)(p) ○ Suspected lack of neutrality of communities or advocacy groups (e.g., CSOs and patient associations) potentially considered as biased groups (p)(q)(s) ○ Considerations in regards the role of lobbies, potentially biased (p) ○ Political correctness perceived as a barrier to investigating by the researcher/s (p) ○ Researchers’ lack of social skills (p)
		• **Comprehensiveness of the strategic plans:** ○ Increased complexity and difficulties in working on co-production frameworks (e)(f)(q) • **Logistic and institutional factors:** ○ Logistic difficulties in attending meetings or interviews (f)(q)(r). ○ Political barriers for obtaining funding (p)(f) ○ Lack of administrative support for researchers (o)(p) ○ Time and costs required for the recruitment (f) ○ Unclear expectations from the point of view of the participants (l)(q) ○ Burdensome demands in the research process, such as the duration and length of questionnaires (p)(r) ○ Participants’ health conditions, limitations, and exacerbations (l) ○ Language barriers (l)

## Data Availability

Concerning availability of data, all tables with excerpts and verbatim quotations extracted from studies sampled are available as complementary materials at https://roderic.uv.es/handle/10550/77735, accessed on 22 November 2021, deposited in the Institutional Repository of the University of Valencia Roderic (roderic.uv.es, accessed on the 22 November 2021).

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
