# Peer review of "Engaging People and Co-Producing Research with Persons and Communities to Foster Person-Centred Care: A Meta-Synthesis"

_ijerph, 2021, doi:10.3390/ijerph182312334_

Round 1
Reviewer 1 Report
The paper is of an overview nature and its value may result from the attempt made to systematize the knowledge. The adopted formula of considerations results in quite general conclusions, although in my opinion many of them are correct. Taking into account the concept of the paper developed by the authors, I believe that such an approach is acceptable. However, it would be valuable to indicate specific recommendations that could "improve co-production processes involving vulnerable groups for supporting communities in fostering their empowerment at collective level".
As far as the editorial side is concerned, the paper needs to be refined, as editorial errors occur (e.g., coloured text highlights, automatic editor messages: "Error! Reference source not found"). My request to the authors is to edit the text carefully.
Author Response
Dear editors and revisor:
Above all, thank you very much for considering our article for publication. Also sincere thanks for your work and your insightful review.
Firstly, and attending to your revision, the Conclusions has been updated
First shortcoming:
However, it would be valuable to indicate specific recommendations that could "improve co-production processes involving vulnerable groups for supporting communities in fostering their empowerment at collective level".
In line with the section on Implications for practice – within the Discussion epigraph – considerations about collective empowerment, structural settings and systemic co-determinants have been summarized and added to the Conclusions. Specifically:
"When designing innovation and/or interventions involving vulnerable communities, socioeconomic and cultural settings and systemic co-determinants should be integrated within the co-production process. Collective empowerment depends on the access to basic services (among others, healthcare) and the material circumstances, equitable societal structures (policies, budget and funds, approach and focus of the measures undertaken by decision-makers) and structural determinants (again, socioeconomic, cultural and environmental conditions co-determining health inequalities at macro level). Co-production processes could be approached for raising the individuals' self-efficacy, skills, and access to services, the advocacy capacities of the community and civil society players, and for supporting long-term changes and promoting healthier policies able to change current vicious cycles of poverty and poor health outcomes. To collaborate with communities and to sustain or, at least, not interfere in the creation of sustainable structures and bonds will result in meaningful empowerment and an enhanced sense of agency of all participants."
Second shortcoming:
As far as the editorial side is concerned, the paper needs to be refined, as editorial errors occur (e.g., colored text highlights, automatic editor messages: "Error! Reference source not found"). My request to the authors is to edit the text carefully.
Secondly, the text has been edited. Changes are marked up using the 'track changes'. Highlights, double spacing between words, Table and Figure numbering and titling, abbreviations, double punctuation and double comas, minor typos, inappropriately bolded words, correction of colours, and inconsistency or hyperlinks to the DOI or online sources, among others, has been addressed. In addition, tables 4 and 5 have been edited for re-adjusting the format to the IJERPH requirements. A non-exhaustive summary is detailed below:
- Highlights: p1, p15
- Double spacing between words: 3, 4, 5, 10, 12, 16, 17, 18, 21
- Line breaks: p4
- Abbreviations: p14 (e.g.,)
- Double punctuation (..) p17; double comma (,,) p18
- Typo (consist on, instead of consist of) p17
- Inappropriately bolded words: 21
- Correction of colours: 21, 22
- Inconsistency of hyperlinks to DOI, articles and other works: Appendix and References
- Tables: Table 5 edited; Table 6 edited and re-adjusted (only format issues covered by the revision)
Additionally, we have conducted a proofreading within the company covering comprehensiveness, grammar and understandability. The revision has been conducted by a proficient English-speaking colleague not involved in the writing of the present article in order to ensure that the readability is appropriate considering the publication and its scope.
Also, errors and inconsistencies were detected within the internal revision in Figure 1. These issues have been addressed: a typo stated that 103 papers were included in the meta-study, being 102 studies the correct number; the text says that 2,729 abstracts were screened after deleting irrelevant and non-related results, but the former Figure 1 did not clarify this step of the revision process appropriately. Its legibility has also been improved, correcting some typos contained in the first version.
Attending to another reviewer's comments, the Figure combining the ladder of participation, social determinants of health, and the typologies of research and intervention has been simplified and moved to Appendix 1.
We would be happy to make any further changes required, and we remain at your disposal.
Kind regards,
Beatriz, Mireia and Maite.

Reviewer 2 Report
I think you need to simplify Figure 1 where you combine the ladder of participation, social determinants of health and the typologies of research and intervention. It could be a supplement in the appendix. The figure is really hard to read, and I do not think they need to include it. I think Figure 3 suffices to explain how they applied these different frameworks to their results.
Author Response
Dear editors and revisor:
Above all, thank you very much for considering our article for publication, and for your work and your insightful review.
First shortcoming:
I think you need to simplify Figure 1 where you combine the ladder of participation, social determinants of health and the typologies of research and intervention. It could be a supplement in the appendix. The Figure is really hard to read, and I do not think they need to include it. I think Figure 3 suffices to explain how they applied these different frameworks to their results.
In order to accomplish the suggestions made in your revision, the Figure combining the ladder of participation, social determinants of health and the typologies of research and intervention has been simplified and, also, moved to Appendix 1 (ie., the former appendix 1 containing the studies comprising the sample has been re-named to Appendix 2):
Typography, distribution and size have also been re-adjusted for improving its readability.
Attending to another reviewer's comments, the current section on Conclusions specifies better the final outcomes of the study in terms of community engagement. Different edition changes have been made. Lastly, errors and inconsistencies detected within the internal revision in Figure 1 have been addressed: a typo stated that 103 papers were included in the meta-study, being 102 studies the correct number; the text says that 2,729 abstracts were screened after deleting irrelevant and non-related results, but the former Figure 1 did not clarify this step of the revision process appropriately. Its legibility has also been improved, correcting some typos contained in the first version.
We would be happy to make any further changes required, and we remain at your disposal.
Kind regards,
Beatriz, Mireia and Maite.
